Morphometric analysis revealed two different Mediterranean horse mackerel (Trachurus mediterraneus) stocks in the Adriatic Sea

Vasapollo Claudio claudio.vasapollo@cnr.it
Institute for Marine Biological Resources and Biotechnology, National Research Council , Ancona , Italy
Yapıcı Sercan
Electronic publication date: 2025 Jan 21
Publication date: 2025
Volume: 13
Electronic Location ID: e18765
Received 2024 Jan 21; Accepted 2024 Dec 4
Copyright: ©2025 Vasapollo
Copyright year: 2025
Copyright holder: Vasapollo
License: This is an open access article distributed under the terms of the Creative Commons Attribution License, which permits unrestricted use, distribution, reproduction and adaptation in any medium and for any purpose provided that it is properly attributed. For attribution, the original author(s), title, publication source (PeerJ) and either DOI or URL of the article must be cited.
License URL: https://creativecommons.org/licenses/by/4.0/

Keywords: Trachurus mediterraneus, Adriatic Sea, Morphometric analysis, Mediterranean Sea, Fish stocks, Mediterranean horse mackerel, Feeding ecology, Discriminant analysis, Population ecology

Funding: The author received no funding for this work.

==============================
Phenotypical differentiation among individuals of Mediterranean horse mackerel Trachurus mediterraneus in the Adriatic Sea was investigated through the analysis of several morphometric characters. Overall, 426 individuals of Mediterranean horse mackerels were sampled from the northern, central and southern Adriatic Sea during the summers of 2012 and 2013. Forty-six morphometric characters were measured for each individual and then compared using multivariate techniques (linear discriminant analysis). Based on the morphometric characteristics, at least two different Mediterranean horse mackerel were identified: one comprising the northern and central Adriatic, and the other formed by individuals from the southern Adriatic basin. The northern and central areas showed stable populations, overlapping both in space and time. The southern area seemed to be more variable over the years, with a low degree of overlapping both in space and time. A possible hypothesis for this, to be further investigated, could be the flow of individuals from the Ionian and Aegean Seas populations through the Otranto Channel. The main differences between the two stocks were associated with the head characters of the fish. In particular, the northern and central Adriatic Sea individuals had shorter and thicker heads than the southern ones. This could be due to different feeding habits: the former mainly feed on small fishes, the latter mainly on euphausiids. A short mouth could reduce the power of suction of bigger preys, while a long mouth could increase the volume of water to be filtered to feed on small planktonic crustaceans. From this study, it becomes clear that the Mediterranean horse mackerel should not be managed as a single stock in the Adriatic Sea as it was evident that at least two morphologically different stocks are present in the basin.

Introduction

Mediterranean horse mackerel Trachurus mediterraneus (Steindachner 1868) is a Carangidae benthopelagic species widely distributed in the Mediterranean and the Black Sea (Fischer, Bauchot & Schneider, 1987). Although this species, and its congeneric ones, represent an important resource in the rest of the Mediterranean Sea (Turan, 2004; Coco et al., 2022), no reliable landing data are available for the Adriatic Sea (Jardas, Santic & Pallaoro, 2004). Estimates of the mean annual landing values for the period 1970—2021 extracted from the GFCM - FAO (General Fisheries Commission for the Mediterranean—Food and Agricultural Organization of the United Nations) capture production database (http://www.fao.org/gfcm/data/capture-production/ar/), assessed the landings of Trachurus spp. (generally, the congeneric species are difficult to be recognized by fishers as the species are similar) in about 5,089 metric tons/year for the whole Adriatic Sea, showing a tendency of reduction in catches starting from the middle of the 1980s. Most of these catches (more than 80%) come from the Italian fisheries that alone reported a mean annual landing value of about 4,274 metric tons year−1 (corresponding to approximately 1.5% of the total Italian fishery landings). Economically, catches of Trachurus spp. from the Italian coasts for the period 2003–2020 have returned a mean profit of approximately 1.5 million euros/year (less than 1% of the total gain of the Italian fisheries; data from the Italian National Institute of Statistic (ISTAT); http://www.istat.it). Notwithstanding the low impact of the landings of this species in the Adriatic Sea, its ecological importance is not negligible, as it represents an intermediate between lower levels (plankton to small pelagic fishes’ eggs and larvae) and top predators (jackfish and tunas) in the trophic web like any other small pelagic fish (Fréon et al., 2005). Although considered as an ancillary species, the Mediterranean horse mackerel is increasing its commercial importance to the point that stock assessments started to be made also for the Adriatic sea (Angelini et al., 2021; Palermino et al., 2023). Following these considerations, scarce are the information on the biology and ecology of the species in the Adriatic Sea (Arneri, 1983; Arneri & Tangerini, 1984; Viette, Giulianini & Ferrero, 1997; Šantić, Jardas & Pallaoro, 2003a; Jardas, Santic & Pallaoro, 2004; Šantić, Radja & Paladin, 2011; Pešić et al., 2012), although a recent new effort have been made to study the acoustic properties of the species for biomass assessment at sea during acoustic surveys in the Adriatic Sea (Palermino et al., 2021; Palermino et al., 2023). Recognizing the extension of the fish stocks and their structures is essential to optimize fishery management and their yield (Begg, Friedland & Pearce, 1999). A misrecognition of the structure of exploited stocks of a species can potentially lead to overfishing and depletion of less productive stocks when multiple stocks are differentially exploited (Begg & Waldman, 1999). Moreover, the degree of exchange between stock members has remained a challenge to fisheries scientists and managers (Campana & Casselman, 1993; Begg, Friedland & Pearce, 1999).

Species identification and stock discrimination are important in conservation of biodiversity of natural resources and fisheries management (Ibañez, Cowx & O’Higgins, 2007). The study of the morphometric characteristics is one of the methods used to identify different fish stocks living in different environmental conditions (Begg & Waldman, 1999; Begg, Friedland & Pearce, 1999; Cadrin, 2000). Morphometrics include the analysis of body shape (or part of the body) and as with meristic characters, its expression is under the simultaneous control of genetic and environmental factors. Contrary to the genetic analysis that proves evolutionary differences between stocks and considered a robust tool for identifying reproductive isolation between stocks, the analysis of the phenotypic characters can provide an indirect way for individuating stock structures indicating prolonged separation of post larval individuals in different environment and/or feeding regimes (Begg, Friedland & Pearce, 1999). The morphometric analysis approach worked well in discriminating Trachurus spp. stocks in the Black Sea, in the Marmara Sea and in the Atlantic regions. For example, Murta (2000) found that T. trachurus (Linnaeus 1758) along the Iberian coasts formed different stocks from the Atlantic coasts of Morocco. Successively, in a wider study comprising northern Atlantic and Mediterranean Sea, Murta, Pinto & Abaunza (2008) showed six different stocks of T. trachurus, three of which were in the Mediterranean Sea. Turan (2004) was able to identify at least three stocks of T. mediterraneus between Black Sea, the Marmara Sea and the Aegean and eastern Mediterranean Sea. Examples are available also for other small pelagic species worldwide such as Scomber australasicus (Cuvier 1832), Engraulis encrasicolus (Linnaeus 1758), Scomber japonicus (Houttuyn 1782) and Sardina pilchardus (Walbaum 1792) (e.g., Tzeng, 2004; Erdoğan, Turan & Koc, 2009; Erguden et al., 2009; Baibai et al., 2012).

Therefore, the purpose of this manuscript is to throw the basis for the assessment of the existence of different stocks of Mediterranean horse mackerel, through a morphometric analysis, in the Adriatic Sea, a semi-enclosed highly productive basin of the central Mediterranean Sea, characterized by different hydrodynamic and bathymetric features between northern, central and southern basin and several promontories and archipelagos that likely act as physical barriers to the expansion of the populations of this species and/or to the mixing of different populations. The present manuscript is the peer reviewed version of a previous preprint available at www.biorxiv.org/content/10.1101/2024.07.23.604807v1.

Material and Methods

A total of 426 individuals of Trachurus mediterraneus were collected along all the Adriatic Sea (Fig. 1). The available data on landings shows that Trachurus spp. yield in the whole Adriatic Sea declined between 1970 and 2020 (GFCM-FAO database (Fig. 2), but see also Angelini et al., 2021). On the Italian side of the Adriatic Sea, landing data clearly show this tendency at least for the period 2005–2020 (ISTAT; Fig. 2) where it is evident that most of the landings came from the south of Italy (in particular, from Apulia region).

Figure 1 Map of the Adriatic Sea where specimen of T. mediterraneus were caught.

The Adriatic Sea was divided into these main areas based on oceanographic characteristics: NA, North Adriatic; CA, Central Adriatic; SW, South-western Adriatic Sea; SE, South-eastern Adriatic Sea.

Figure 2 Trend of the total landings of Trachurus spp. in the Adriatic Sea and the rest of the Mediterranean Sea (Northern, Central, Southern and Total Italy data from the Italian National Institute of Statistics (ISTAT); Total Adriatic (both western and eastern) from General Fisheries Commission for the Mediterranean–Food and Agricultural Organization of the United Nations (GFCM –FAO)).

The basin was ideally divided into 3 main areas (according to the mean cyclonic circulations and bathymetry; Artegiani et al., 1997): (1) Northern Adriatic (NA) from the Gulf of Trieste to the promontory of Ancona and with an average depth of 35 m; (2) Central Adriatic (CA) from the promontory of Ancona to the promontory of the Gargano mountain and an average depth of 140 m, with the two Pomo depressions reaching 260 m; (3) and southern Adriatic with an average depth >500 m and a depression of >1200 m, subdivided into South-Western (SW) and South-Eastern (SE) Adriatic. Specimens were obtained by means of pelagic trawls (Table 1) made during two echo-survey cruises in the framework of the MEDIAS project (MEDiterranean International Acoustic Survey; Leonori, Tičina & De Felice, 2011) held in 2012 and 2013 in the northern-central (September) and southern (July–August) Adriatic Sea corresponding to the reproductive period of this species (Arneri, 1983; Viette, Giulianini & Ferrero, 1997).

According to Reist’s (1985) recommendation that at least 25 individuals should be used for morphological analyses, a range of 24 to 68 individuals were collected in 2012 and 50 to 74 in 2013 of approximately the same size ranges and in any case adult individuals (Table 1). The individuals were kept deep-frozen (−20 °C) for transport to the laboratory, where several landmarks were pinned to the fishes before taking photographs (Canon PowerShot SX260 HS, 50 mm normal lens to avoid distortion of the image). By means of the image free software ImageJ v.1.54 g (Abramoff, Magalhaes & Ram, 2004), a truss-network was built accounting for a total of 43 lengths, while three measures of width were taken by means of a digital caliper (Fig. 3).

Table 1 General information on the areas of sampling and length of T. mediterraneus individuals.

Areas of sampling, number of hauls from which specimen of T. mediterraneus come, number of individuals per area and year, number of females (F), males (M) and immature (I) individuals per area and range of the standard length per area. The grand mean of the standard length was 188.6 mm.

Area/Year	Individuals	No. Hauls	F	M	I	Standard Length Range (mm)	
NA12	68	4	40	25	3	128.5–237.5	
NA13	72	7	43	29	–	134.6–243.6	
CA12	50	4	26	24	–	160.8–231.5	
CA13	74	5	41	30	3	146.5–245.4	
SW12	36	4	19	17	–	204.8–293.2	
SW13	50	4	22	28	–	152.8–293.1	
SE12	24	2	13	11	–	135.6–253.5	
SE13	52	5	24	28	–	136.9–220.6	
Total	426	35	228	192	6	Average Standard Length = 188.6 mm	

Figure 3 Locations of the 19 pins and 43 lengths of the truss network of T. mediterraneus used for the multivariate analysis.

IW, interorbital width; CW, central width; PW, peduncle width; C_D, eye diameter; 1_G, head length; E_F, pectoral fin length; 14_B, ventral fin length; 3_A, first dorsal fin height.

To remove the size dependent effect of each morphometric measure the Elliott, Haskard & Koslow (1985) formula was applied: Madj=MSLs/SLob

where M is the original morphometric measurement, Madj the size adjusted measurement, SLs the overall mean of standard length for all fish from all samples (188.6 mm) and SLo the standard length of each single individual. The parameter b was estimated for each character as the slope of the regression of log M vs log SLo. Correlation coefficients between transformed measurements and standard length were calculated to check if the data transformation was effective in removing the effect size. If any correlation among variables persisted after transformation, one of the variables was maintained as a proxy for the others. The effect of sex on the measurements was not tested since Mediterranean horse mackerel does not show any sexual dimorphism (Jardas, Santic & Pallaoro, 2004; Turan, 2004; Kutsyn, 2021).

After data standardization, an un-weighted pair-group method using arithmetic averages (UPGMA) cluster analysis of the Euclidean distances of the eight groups (each single area for each single year) of individuals was performed to assess the presence of clusters of fish aggregations (Legendre & Legendre, 2012; Borcard, Gillet & Legendre, 2018). Once obtained the clusters, a linear discriminant analysis (LDA) on standardized data was performed to identify the combination of variables that best separated T. mediterraneus samples. It proceeds in two steps: (1) one tests for differences in the morphometric variables among the predefined groups; (2) if the test supports the alternative hypothesis of significant differences among groups, the analysis proceeds to find the linear combinations (called discriminant functions) of the variables (standardized coefficients) that best discriminate among groups (Legendre & Legendre, 2012). After cross-validation (jackknife), each fish individual is allocated to the group with the nearest centroid, and the proportion of individuals allocated to each group is calculated after production of a so-called confusion matrix. The proportion of correct allocation is taken as a measure of the integrity of that group. To test the assumption of multivariate homogeneity of within-group covariance matrices a permutational test has been applied (Borcard, Gillet & Legendre, 2018). To estimate the significance of the differences among groups, a Permutational Multivariate Analysis of Variance (PERMANOVA; Anderson, 2001) was performed between each pair of groups of a Euclidean matrix built with the Elliot’s transformed morphometric measurements.

Statistical analyses were performed with R packages stats v.4.4 (hclust function; R Core Team, 2024, MASS v.7.3 (lda function; Venables & Ripley, 2002) and vegan v.2.6 (adonis2, betadisper, permutest functions; Oksanen et al., 2024).

Results

A total of 426 individuals (average standard length = 188.6 cm; range size from 128.5 cm to 293.2 cm) have been collected for morphometric analysis during 2012 and 2013 cruises in the Adriatic Sea from a total of 35 hauls (Table 1). Most of the morphometric variables measured were highly size dependent but the Elliot’s transformation was effective for almost all the variables (Table S1). Notwithstanding, seven variables (1_8, 8_9, 4_12, 4_11, 5_10, 11_6 and 3_13) still showed collinearity, thus, to reduce their redundant effect on the analysis, they were dropped from the list when the correlations were >0.90 (Table S1), overall reducing the number of morphometric variables to 36.

The cluster analysis showed the division of individuals in mainly two groups based on the geographical separation between south and north-central Adriatic (Fig. 4). Based on the cluster analysis and following the hypothesis of the presence of different horse mackerel stocks in the basin, the within-group analysis did not show any difference (Table 2). On the other hand, PERMANOVA clearly showed a difference between areas (Table 2), confirming the separation of the stocks in different areas of the Adriatic Sea.

Figure 4 Dendrogram of the cluster analysis (UPGMA) for the centroids of the groups based on Euclidean distances of morphometric data.

Cophenetic correlation = 0.93. NA, North Adriatic; CA, Central Adriatic; SW, South-western Adriatic Sea; SE, South-eastern Adriatic Sea.

Table 2 PERMUTEST and PERMANOVA results.

Results of the PERMUTEST, to test the within-group multivariate homogeneity, and of the PERMANOVA, to test the between-groups differences. The number of permutations for each test was set to 9999.

PERMUTEST	PERMANOVA	
	df	F	p		df	F	p	
Area Year	7	1.3	0.2667	Area Year	7	7.1	0.0001	
Residuals	418			Residuals	418			
				Total	425			
		No. Perm = 9999			No. Perm = 9999	

Figure 5 Plot of the centroids of the sample groups in the space defined by the linear discriminant analysis.

Both axes explained together 72.5% of the total between-group variability (first axis 54.9%, second axis 17.6%). Dots represent centroids of the samples per area per year. The radius of the ellipses around each centroid corresponds to one standard deviation of the Euclidean distances from each individual to its group centroid. Arrows represent the contribution of morphometric variables to the canonical functions (standardized coefficients). Only the most correlated vectors (≥—1—) are reported. NA, North Adriatic; CA, Central Adriatic; SW, South-western Adriatic Sea; SE, South-eastern Adriatic Sea.

The LDA analysis plot showed a clear separation between the northern/central areas respect to the southern area, suggesting the existence of at least two stocks of Mediterranean horse mackerel (Fig. 5). Moreover, the northern/central area groups maintained a temporal stability between the two years being all groups close each other, while, on the other hand, the southern area groups were scattered showing temporal instability and suggesting the further presence of ephemeral populations changing each year. The first two discriminant axis produced by LDA explained most of the total between-group variability in the dataset (72.5%; LD axis1 = 54.9% and LD axis 2 = 17.6%), showing clear between-groups differentiation. The confusion matrix produced by the discriminant analysis (Table 3) showed that the cross-validated percentages of perfect classification of the individuals into the corresponding groups was 51.4%. The higher value of good correspondence was in the area SE13 (67.3%) while the lowest value was in CA12 (34%). To make the plot as much clear as possible, standardized coefficients were selected based on the absolute values (≥—1—) to highlight those variables that mostly driven the spatial configuration (Fig. 5). Two variables were strongly associated to the first most explanative axis corresponding to the length of the head and to the distance from the upper portion of the head until the operculum opening of the individuals (1_G and 2_G). On the other hand, some of the variables corresponding to the peduncle of the caudal fin and of the anterior part of the body were associated to the vertical axis (Fig. 5).

Table 3 Confusion matrix produced by the linear discriminant analysis.

Numbers (and percentages) represent T. mediterraneus individuals correctly classified into their original group after cross-validation of the discriminant analysis of the morphometric data. Rows are the original sample group and columns the reallocation group after jackknife cross-validation. r-SUM, row sum; c-SUM, column sum.

	NA12	NA13	CA12	CA13	SW12	SW13	SE12	SE13	r-SUM	
NA12	40 (58.8%)	7	13	2	0	4	1	1	68	
NA13	10	35 (48.6%)	4	12	2	7	1	1	72	
CA12	22	2	17 (34%)	8	0	1	0	0	50	
CA13	4	19	5	34 (46%)	1	11	0	0	74	
SW12	1	3	1	1	18 (50%)	0	6	6	36	
SW13	0	8	2	12	0	26 (52%)	0	2	50	
SE12	0	0	0	1	4	0	14 (58.3%)	5	24	
SE13	0	0	0	3	7	3	4	35 (67.3%)	52	
c-SUM	77	74	42	73	32	52	26	50	426	
						Percentage of Correct Classification = 51.4%	

Discussion

Phenotypical differences do not provide a direct clue of genetic isolation between stocks, although they can show a prolonged separation of post larval fish exposed to different environmental regimes (Begg & Waldman, 1999). With this premise in mind and considering that morphometrical analysis is just one of the instruments that fishery researchers can adopt to distinguish different stocks, the actual morphometric characters of Mediterranean horse mackerel showed significant phenotypic distinctness between north-central and southern Adriatic Sea. The present results suggest the existence of at least two different stocks based on phenology, even if the southern groups are likely as different each other in both years as respect to the north-central population. Even though, a certain degree of mixing is plausible above all between the two populations at the border of the two areas CA and SW (at least for one year), but in any case, it seems not the rule but rather an exception, although new samples from different years are needed that potentially could confirm or not this result. The individuals, from the north and the central Adriatic, were well mixed each other in both years. Under these circumstances, it is plausible to indicate the northern and central Adriatic populations as (at least morphologically) closed and self-recruiting stock because of their temporal stability. A clear idea of this hypothesis could come from a further genetic study confirming (hopefully) the segregation of this population respect to the southern Adriatic and (potentially) even from the rest of the Mediterranean. In fact, the southern individuals seemed to be quite different each other in both years. A possible explanation could be the influence of Mediterranean populations of T. mediterraneus entering the Adriatic Sea toward the Otranto Channel, thus favoring the constant flow into the southern Adriatic of new individuals each year. It is likely that the conditions (both environmental and feeding conditions) in the southern Adriatic are quite similar to the Ionian and Aegean Sea. In effect, the south Adriatic has open sea water mass characteristics as the northern Ionian Sea and the surface transport of water masses into the southern Adriatic occur from the eastern Ionian Sea (Vilibić et al., 2012) likely transporting individuals from Greek populations. This hypothesis should be proven by studying both southern Italian Ionian and Greek populations of Mediterranean horse mackerel both genetically and morphologically.

The separation among stocks might be due to the considerable differences in environmental conditions such as temperature, salinity and food availability between the three regions here considered (Artegiani et al., 1997). In fact, it is well known that fish exhibit morphological variation induced by environment (Cadrin, 2000). One of the clues that lead to the conclusion of individuals living in different habitats came from the discriminant analysis that showed that morphometric differentiation among samples was largely due to differences in the head features of fish. Also, Turan (2004) found the same Mediterranean horse mackerel head differences along the Turkish coasts demonstrating that the most important leading variables in discriminating among stocks are associated to the prey selection in the different habitats, and, as reported by Gatz (1979), the relative head length would be linked to prey size. From the present results emerged that the northern and central western Adriatic individuals were characterized by shorter but thicker head than the southern populations. This is not negligible since the diets between northern and southern populations are different enough to further support the distinction between Adriatic populations. Šantić, Jardas & Pallaoro (2003b) observed that the main preys of Mediterranean horse mackerel in the central eastern Adriatic Sea (deeper than western one) were plankton euphausiids, representing the 50% of the preys found in the stomachs analyzed. The rest of the food was represented by teleosts (as secondary food) and other planktonic crustaceans (as occasional food). Euphausiisds are representative of deep and open waters, and, in fact, they are present in the southern and deepest parts of central (eastern) Adriatic while in the north this planktonic group is not present (Šantić, Jardas & Pallaoro, 2003b). It is true that the prey size preference changes with fish size class. For example, Šantić, Jardas & Pallaoro (2003b) found that individuals under 28 cm total length prefer the planktonic crustaceans than small teleosts while Yankova, Raykov & Frateva (2008) found a preference of mysid crustaceans in individuals less than 11 cm length and a preference of fishes (anchovy and sprats) in longer Mediterranean horse mackerel individuals from the Bulgarian Black Sea. In the present study, individuals from north to south did not have lengths lower than 15 cm, and the size class ranges here analyzed are comparable among the areas and years and the differences found between the head lengths corresponded to effective phenotypical differences due to different habits. Therefore, the differences in the head measures could be interpreted in respect to the kind of prey a Trachurus population feed on. For example, the northern and central populations potentially feeding mainly in small teleosts may be advantaged by having a short and thick mouth to facilitate the suction of the preys, while on the contrary, a long mouth could facilitate filtering a large volume of water to feed on plankton euphausiids. Regarding the morphometric characteristics associated with the second discriminant axis it is more difficult to explain them. Notwithstanding, it is recognized that the kind of feeding habits promote changes in the body morphology of fishes (Blake, 2004). For example, the measure 2_13 (corresponding to the 2_12 in Turan, 2004), as well as the peduncle measures, are likely correlated to the swimming optimal functional design of the fish body. To maximize thrust and minimize drag a streamlined body is more effective, as those of the northern Adriatic individuals that need a rapid movement to catch fishes to feed on. The same solution is needed also to escape from predators (e.g., tunas) and this could explain why also individuals from the southern Adriatic (at least for the 2012) share these characteristics with the northern ones but could potentially due also to other intrinsic and/or extrinsic reasons. Interestingly, individuals from southern and central Adriatic caught in 2013, showed an opposite body shape respect the 2012 individuals. This point should be investigated further. Nevertheless, these considerations should be viewed as speculative mostly because the second discriminant axis explained variance is quite low and thus, they should be taken with precaution.

Conclusion

It is certain that for a reliable identification of several stocks firstly detected by a single analysis, a multi-methodological approach (the so called holistic approach) should be used (Begg & Waldman, 1999). The results of the present work should be compared and/or confirmed by analyzing data obtained from other phenotypic characteristics and genetic studies even from different sites (and possibly from the eastern Adriatic Sea). For example, another approach might be the study of otolith morphometry and chemistry that is increasingly used to discriminate between fish stocks (Campana & Casselman, 1993; Turan, 2006; Jemaa et al., 2015). If the holistic approach will confirm the existence of two (or more) different T. mediterraneus stocks, this could have consequences regarding their management. So far, the Mediterranean horse mackerel in the Adriatic Sea has been treated as a single stock (Angelini et al., 2021), but the present morphometric results showed that at least two stocks are present in the Adriatic waters. Unfortunately, no morphometric data is available from the northern-eastern and central-eastern Adriatic basin, and this is a pity, but already the results from the present study suggest that from a management point of view, the existence of different stocks could have huge consequences, above all if such population structure persists over time. If persistence occurred, it could be important to manage in different ways the different stocks because any depletion in one of them is unlikely to be compensated for by immigration from other stocks, at least in a short time. Thus, for accurate assessment of the state of a stock, not only the boundaries but also the mixing levels between stocks should be considered as well as the state of the environment conditions (above all the foraging species depletion; Turan, 2004; Murta, Pinto & Abaunza, 2008).

Supplemental Information

Table S1 Correlation coefficients between morphometric characters

Correlation coefficients between morphometric characters, before and after the removal of the size effect, are respectively shown below and above the diagonal. Correlation values higher or equal to 0.90 are in bold.

Data S1 Raw data of fish morphometric measurements

Each row corresponds to an individual fish.

The author would like to thank the crew of the N/O “Dallaporta” for assistance during the surveys, and the MEDIAS project group of the IRBIM-CNR, especially Giovanni Canduci for the support during the hauls and the project coordinator Iole Leonori. MEDIAS is one of the mandatory surveys of the European Data Collection Framework and a big thanks goes also to the current Italian coordinator of the Italian program, Enrico Arneri. I also want to thank the three reviewers for their precious suggestions that greatly ameliorated the quality of the original manuscript.

Additional Information and Declarations

Competing Interests

Author Contributions

Data Availability

Claudio Vasapollo is an Academic Editor for PeerJ.

Claudio Vasapollo conceived and designed the experiments, performed the experiments, analyzed the data, prepared figures and/or tables, authored or reviewed drafts of the article, and approved the final draft.

The following information was supplied regarding data availability:

The raw measurements available in the Supplementary File.

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
