# Peer review of "Morphometric analysis revealed two different Mediterranean horse mackerel (Trachurus mediterraneus) stocks in the Adriatic Sea"

_PeerJ, doi:10.7717/peerj.18765_

## Round 0.1 · original submission · Major Revisions

Dear Dr. Vasapollo

You can find the comments and suggestions of the expert reviewers in the attached reports. As you will see, expert reviewers have pointed out the critical errors (e.g. lack of relevant literature, confusion about the terms you use, and deficiencies in your hypotheses and methodologies). Consequently, a major revision is needed for your article.

I request you check and correct the manuscript based on the reports.

Sincerely

·

Basic reporting

No comment

Experimental design

No comment

Validity of the findings

No comment

Additional comments

In line 67-69, you can also add:
Species identification and population discrimination are important in conservation of biodiversity of natural resources, and fisheries management (Iba_nez et al., 2007).

Iba_nez, A. L.; Cowz, I. G.; O’Higgins, P., 2007: Geometric morphometric analysis of fish scales for identifying genera, species, and local populations within the Mugilidae. Can. J. Fis. Aquatic. Sci. 64, 1091–1100.

In line 191, what kind of data is required, please specify it.

In line 269, “suggest that from a management point of view” this is an important point of the conclusion. Please add more and detail suggestion of your paper about the management of this species related with the legistlations in the region and also it will be better conclusion to mention about how to manage two stocks in two areas close to each other.

Reviewer 2 ·

Basic reporting

The abstract presents an interesting study on the phenotypic differentiation of Trachurus mediterraneus stocks in the Adriatic Sea, but significant revisions are needed to enhance the scientific robustness and relevance. The work lacks references to important studies in the field, and it is crucial to include more recent literature, such as the paper "Morphometric Analyses of Phenotypic Plasticity in Habitat Use in Two Caspian Sea Mullets" by Shima Bakhshalizadeh et al., 2022, along with other key studies provided in the attached PDF. These additions would strengthen the theoretical framework and position the research more effectively within current scientific discourse.

Experimental design

The conclusion that two distinct stocks exist based solely on morphometric analysis is oversimplified. Morphometric differences can indicate phenotypic variation, but they are not sufficient to confirm stock separation. The authors need to adopt a more critical approach, recognizing the limitations of relying only on morphometrics, and consider integrating genetic or ecological data to better support their conclusions. The ecological hypothesis suggesting a link between morphological differences and feeding habits is plausible but requires more substantiation. Citing ecological or trophic studies specific to Trachurus mediterraneus in the Adriatic would add depth to this argument, as it currently lacks sufficient evidence. Additionally, the abstract mentions multivariate techniques but does not provide enough detail on the specific morphometric variables analyzed. Greater clarity on the selection criteria and the rationale behind the chosen variables is necessary to ensure methodological transparency.

Validity of the findings

The claim that northern and central Adriatic populations are more stable compared to southern ones is intriguing, yet the abstract fails to offer enough context on the temporal and spatial dimensions of these observations. A more detailed discussion of these factors, including potential environmental influences such as the flow of individuals from the Ionian and Aegean Seas, would be beneficial. Finally, the authors should broaden the discussion on the use of morphometric analyses for stock identification in general, addressing both its strengths and limitations, particularly in light of the recent findings from the Caspian Sea mullet study.

Additional comments

See pdf attached

Annotated reviews are not available for download in order to protect the identity of reviewers who chose to remain anonymous.

Reviewer 3 ·

Basic reporting

The work entitled "Preliminary results from a morphometric analysis: how many Mediterranean horse mackerel (Trachurus mediterraneus) stocks are there in the Adriatic Sea?" analyzes the morphology of T. mediterraneus in the Adriatic seas. The topic of this work is interesting, but the work presents several problems. First of all, the context described in the introduction describes the morphological analysis as the tool to be able to discriminate the stock units. In reality, morphological analyses are one of the elements (e.g. see https://doi.org/10.1016/S0165-7836(99)00065-X, https://doi.org/10.1093/icesjms/fsw188) for the identification of stock units, together with other elements such as biological characteristics (e.g. growth, reproduction), genetic analyses, fishing ground etc. that contribute to the definition of stock units. So it seems that the author confuses the concept of population and stock unit. Regarding the sample the author does not take into account and does not test the possible differences between sexes and between juveniles/adults. This last aspect could have influenced the results and their interpretation (see specific comments). Regarding the statistical methodologies applied they seem adequate.

Experimental design

see specific comment on the pdf

Validity of the findings

see specific comment on the pdf

Additional comments

given the interest and statistical approach that the author has used I advise the author to resolve the issues and resubmit the work

Annotated reviews are not available for download in order to protect the identity of reviewers who chose to remain anonymous.

---

## Round 0.2 · accepted · Accept

Dear Dr. Vasapollo,

I thank you for making the corrections and changes requested by the reviewers. I read and checked your valuable article carefully and am happy to inform you that the article has been accepted for publication in PeerJ.

·

Basic reporting

no comment

Experimental design

no comment

Validity of the findings

no comment

Additional comments

The suggestions have been done.

Reviewer 2 ·

Basic reporting

Dear authors, the suggestions and corrections have been appropriately followed. I now recommend accepting the work in its current form.

Experimental design

No further comments

Validity of the findings

No further comments

Additional comments

No further comments